# SCALING PROPERTIES FOR ARTIFICIAL NEURAL NETWORK MODELS OF THE *C. elegans* NERVOUS SYSTEM

## ABSTRACT

The nematode worm *C. elegans* provides a unique opportunity for exploring intrinsic neural dynamics, given its transparency and well-characterized nervous system. This study delves into the scaling properties vital for self-supervised neural activity prediction, focusing exclusively on neural data and excluding behavioral aspects. We investigate how predictive accuracy, assessed using mean squared error (MSE), correlates with the volume of training data and examine factors such as neuron number, recording duration, and dataset diversity. Furthermore, we analyze how these scaling properties interact with different aspects of artificial neural network (ANN) models, including size, architecture, and hyperparameters. Our findings reveal a logarithmic reduction in MSE with increased training data, consistent across diverse datasets. We also observe nonlinear MSE changes with varying ANN sizes. These insights emphasize the need for advanced imaging tools to extend our understanding of mesoscale nervous systems and inform the development of precise ANN models for neural dynamics, impacting both neuroscience and AI.

## 1 INTRODUCTION AND RELATED WORK

### 1.1 INTRODUCTION

Exploring neural system dynamics is crucial in neuroscience and artificial intelligence (AI). This intersection has spurred the evolution of artificial neural network (ANN) models, inspired by biological neural systems. ANNs offer the potential to emulate diverse animal behaviors, providing advantages like detailed specification, causal manipulability, and increasing analytical accessibility, reflecting key aspects of biological nervous systems (Yamins & DiCarlo, 2016; Yamins et al., 2014). The nematode *Caenorhabditis elegans* (*C. elegans*) is an exemplary model in this context, offering a valuable platform for comparing real and artificial neural dynamics.

### 1.2 *C. elegans* AS A MODEL SYSTEM

*C. elegans* is an excellent model organism for neural dynamics research due to its well-mapped connectome and capabilities for non-invasive neuronal activity tracking via advanced imaging techniques (Leifer et al., 2011; Nguyen et al., 2016). The organism's compact size, transparency, and well-annotated genome simplify intricate optical measurements and deep insights into neural activity. NeuroPAL, a multicolor atlas, allows precise in vivo neuron identification, enhancing the capabilities for measurement and analysis of the *C. elegans* nervous system (Yemini et al., 2021).

### 1.3 SELF-SUPERVISED NEURAL ACTIVITY PREDICTION

Predicting future neural activity based on historical data is a growing field, with advancements in models like LSTMs demonstrating success in complex mammals (Pandarinath et al., 2018). In *C. elegans*, the simplified behavioral repertoire and consistent biology offer a unique setting for in-depth model analysis. Self-supervised learning, predicting future states from intrinsic neural patterns, reduces dependence on behaviorally annotated data. While acknowledging the importance

of behavior in neural dynamics, our study concentrates on the inherent predictability within neural activity, exploring how neural dynamics can be predicted without direct behavioral reference, similar to how large language models (LLMs) uncover intricate structures in language data (Radford et al., 2019; Brown et al., 2020).

### 1.4 NEURAL NETWORK SCALING PROPERTIES

Research into ANNs' scaling properties has shown that improvements in model size, data volume, and computational resources significantly enhance performance (Kaplan et al., 2020b; Hoffmann et al., 2022). The relationship between data size and model capacity is critical in optimizing model performance. However, this relationship in the context of predicting neural dynamics in biological organisms like *C. elegans* is not well-explored. Our study aims to fill this gap by analyzing the impact of data volume, model architecture, and size on ANN performance in neural activity prediction in *C. elegans*. These insights are crucial for optimizing experimental and modeling strategies in neuroscience, contributing to the development of more accurate predictive models for biological nervous systems.

## 2 METHODS

### 2.1 NEURAL ACTIVITY DATA

**Data sources**. We leveraged eight (8) open-source datasets (Atanas et al., 2023; Randi et al., 2022; Yemini et al., 2021; Uzel et al., 2022; Kaplan et al., 2020a; Skora et al., 2018; Nichols et al., 2017; Kato et al., 2015) detailing neural activity in *C. elegans* (see Table 1 for download sources and associated publications). These datasets, each recorded under varying experimental conditions, quantify neural activity through the measurement of calcium fluorescence changes in specific subsets of the worm's 302 neurons. Conditions ranged from freely moving (Atanas et al., 2023), immobilized (Uzel et al., 2022), and asleep (Nichols et al., 2017) states, to optogenetically stimulated scenarios (Randi et al., 2022). These differing conditions were not considered during our modeling. Refer to Figure 1 for a summary of the datasets.

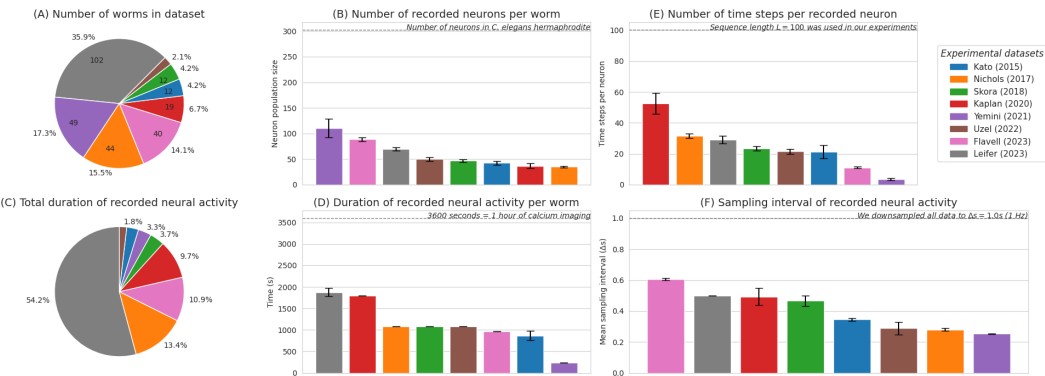

Figure 1: Comprehensive overview of eight open-source *C. elegans* neural datasets utilized in this study: (A) Pie chart showing the distribution of the number of worms across the datasets. (B) Bar graph depicting the average number of neurons recorded per worm for each dataset, with a dashed horizontal line marking the total neuron count in a *C. elegans* hermaphrodite. (C) Pie chart illustrating the total duration of recorded neural activity aggregated from all worms within each dataset. (D) Bar graph representing the average duration of neural activity recorded per worm, highlighted with a dashed line at the 1-hour mark. (E) Bar graph of the number of time steps per neuron recorded, with a dashed line denoting the sequence length used in this study's experiments. (F) Bar graph indicating the mean sampling interval of neural activity recording for each dataset, with a dashed line indicating the standardized sampling interval used for data downsampling. This figure encapsulates the heterogeneity and processing steps taken to standardize the datasets, ensuring comparability for subsequent analyses.

**Standard data format**. Each dataset, represented as $\mathcal{D}$, includes individual recordings from $N$ worms, each consisting of neural activity and a mask indicating the subset of the 302 neurons that were measured and labelled. This mask ensures models are trained only on neural activity recorded from labelled neurons (Figure 3A illustrates the dataset format). Specifically:

$$\mathcal{D} = \{\mathbf{X}^1, \mathbf{X}^2, \ldots, \mathbf{X}^N\} \times \{\mathbf{y}^1, \mathbf{y}^2, \ldots, \mathbf{y}^N\}, \quad N = |\mathcal{D}|$$

Here, each worm $k$ is symbolized by a matrix $\mathbf{X}^k \in \mathbb{R}^{302 \times T_k}$ and a binary column vector $\mathbf{y}^k \in \mathbb{R}^{302}$, specifying which neurons were recorded and are labeled, ordered alphabetically by the canonical names of the neurons. [1] Each row $\mathbf{x}_i^k$ of the matrix $\mathbf{X}^k$ contains the time series of neural activity for the $i^{th}$ neuron over $T_k$ time steps, with rows ordered analogously to $\mathbf{y}^k$.

**Preprocessing.** The data, denoted as $\mathbf{X}^k$, is processed from the original raw data (see Table 1 for dataset details about the individual experimental datasets). To accommodate the unique neural dynamics of C. elegans and maintain the causality of the time series signal, we apply an exponential kernel smoothing method with a smoothing parameter $\alpha = 0.1$. This method ensures that only current and past data points are used for computing the smoothed value, thus preserving the temporal causality in the data.

Considering the variability in experimental imaging sampling rates across different datasets, we standardized the data to a uniform timestep, $\Delta t$. Given the original datasets' sampling rates, we set $\Delta t = 1.0$ second (equivalent to 1 Hz), thereby ensuring that the adjustment process results in consistent downsampling without introducing artificial data points through interpolation.

**Train-test split.** For each worm's neural activity data matrix $\mathbf{X}^k$, we performed a temporal split to create a training set $\mathbf{X}_{\text{train}}^k$ and a testing set $\mathbf{X}_{\text{test}}^k$. A balanced 50:50 split was adopted, allocating the first half of the neural activity recording to the training set and the second half to the testing set. This approach was chosen to ensure that both training and testing datasets are representative of the entire range of neural activities observed.

**Amount of Data.** The scaling of training data is central to our investigation of self-supervised learning models' performance in predicting the next timestep of neural activity. Our collective dataset, denoted by $\mathcal{D}$, comprises 284 worms sourced from eight distinct experimental worm datasets. We methodically scaled up the amount of training data available to the models by varying the size of the dataset $\mathcal{D}_n$, where $n$ indicates the cumulative number of worms included.

$$\mathcal{D}_n = \bigcup_{i=1}^{m} \mathcal{D}_i^{(k_i)}$$

Here, $\mathcal{D}_i^{(k_i)}$ represents the $k_i$ number of worms sampled from the $i^{th}$ experimental dataset, and $m$ is the total number of experimental datasets. The sampling process is akin to a multinomial distribution where the probabilities correspond to the proportion of available worms from each experimental dataset. The combined datasets $\mathcal{D}_n$ thus represent a diverse cross-section of neural activities encompassing variations in experimental conditions.

**Mixed Datasets.** To create mixed datasets, we employed a random sampling technique from the pool of all available worms, producing datasets that encapsulate the diversity stemming from the different experimental paradigms of the original datasets. This randomness is formalized by the multinomial sampling given by:

$$\mathcal{D}_n = \text{Multinomial}(n; p_1, p_2, \ldots, p_m)$$

where $p_i$ reflects the relative contribution of the $i^{th}$ dataset to the pool, based on the number of worms it contains. The result is a series of mixed datasets, each with a unique composition of worms, yet collectively spanning the full range of neural dynamics present in the collective data.

---

[1] https://www.wormatlas.org/NeuronNames.htm

**Individual Datasets.** We can also generate subsets from a single experimental dataset by restricting our random sampling to that specific set, thereby creating increasingly larger subsets and maintaining consistency within the experimental context.

The experimental datasets contributing to $\mathcal{D}$ are denoted as: Kato ($\mathcal{D}_1$), Nichols ($\mathcal{D}_2$), Skora ($\mathcal{D}_3$), Uzel ($\mathcal{D}_4$), Yemini ($\mathcal{D}_5$), Kaplan ($\mathcal{D}_6$), Flavell ($\mathcal{D}_7$), and Leifer ($\mathcal{D}_8$). The scaling experiment then tests the models across discrete dataset sizes ranging from $\mathcal{D}_1$ to $\mathcal{D}_{284}$, with the size and composition of each dataset determined by the multinomial sampling process.

$$\mathcal{D}_{\text{size}} = \bigcup_{i=1}^{8} \mathcal{D}_i^{(k_i)} \text{ such that } \sum_{i=1}^{8} k_i = \text{size}$$

For each discrete dataset size desired, we perform multiple experiments with different random seeds, where we uniformly and randomly select an assignment from all possible assignments that yield the desired size.

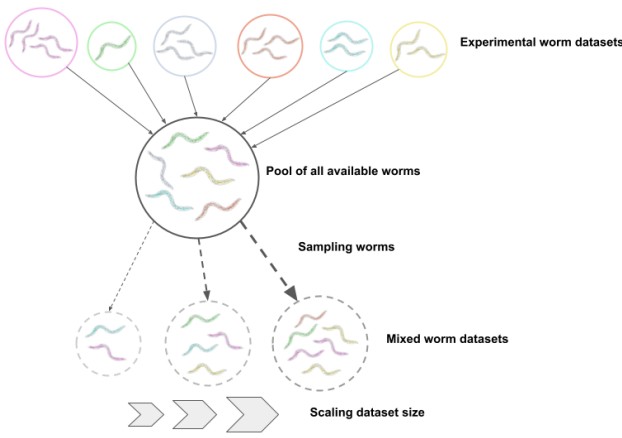

Figure 2: Schematic representation of the method used for creating mixed worm datasets by sampling from the pool of all available worms across different experimental datasets. The process allows for systematic scaling of dataset size to investigate the effect on model performance.

## 2.2 MODEL STRUCTURE

**Model architectures.** Our study utilizes three distinct classes of neural networks to harness different inductive biases for the prediction of future neural activity in *C. elegans*. These include Long-Short Term Memory (LSTM) networks, Transformer networks, and Feed-Forward networks. These architectures were chosen to represent a fundamental set of mechanisms—recurrence, attention, and feed-forward processing—allowing us to assess the impact of structural and mechanistic differences on the task at hand.

**Shared model structure.** Each architecture is implemented within a common structural framework comprising an embedding module, a core processing module, and a linear output mapping to enable a consistent training and evaluation procedure. This shared structure allows for direct comparison of the architectures by substituting only the core module, thereby isolating the effects of architectural differences (Figure 3B).

1. *Embedding:* The initial embedding layer projects the 302-dimensional input representing the neural state space to a higher-dimensional latent space with $H$ hidden units. This transformation is applied through a nonlinear ReLU activation function. Optionally, layer normalization may be applied post-activation to stabilize the learning process. Notably, the Transformer model employs positional encoding in its embedding to account for the sequence order, absent in the other architectures.

2. *Core:* The core module is architecture-specific and constitutes the primary computational engine of the model. It consists of a single layer to maintain simplicity and interpretability, which is particularly important when relating model weights to biological neural networks. For the LSTM and Transformer architectures, the core is inherently causal, ensuring that predictions are based solely on past and present data. We use an encoder layer with 4 attention heads for the Transformer core. The Feed-Forward model lacks access to temporal context beyond the current step, essentially restricting it to feature regression, thus providing a baseline for the importance of temporal information in prediction.

3. *Output Mapping:* The final component of the model is a linear projection from the latent space back to the original neural state space dimensionality. This mapping generates the predicted future neural activity, denoted mathematically as $\hat{\mathbf{X}} = f_{ho}(\mathbf{Z})$, where $f_{ho}$ is the linear transformation from hidden to output space.

**Prediction Task.** Central to our investigation is the one-step prediction task where the model predicts the neural activity at time $t$ based on the activity at time $t - 1$. This task mimics the self-supervised sequence-to-sequence prediction paradigm, with the focus on immediate future state anticipation. The models are trained to minimize the difference between the predicted and actual neural activity, employing Mean Squared Error (MSE) as the loss metric.

**Causal Predictions and Temporal Memory.** In line with the self-supervised learning framework, our models are tasked with making causal predictions, where future predictions do not rely on future inputs. The LSTM model is causal by definition. The Transformer model uses a causal self-attention mask. The Linear model also respects causality as it processes each time point independently. The ability of the models to perform auto-regressive prediction is qualitatively assessed in Figure S(...). This probes their capacity to leverage internal states learned from training on the self-supervised one-step prediction objective for generating sequences of predictions.

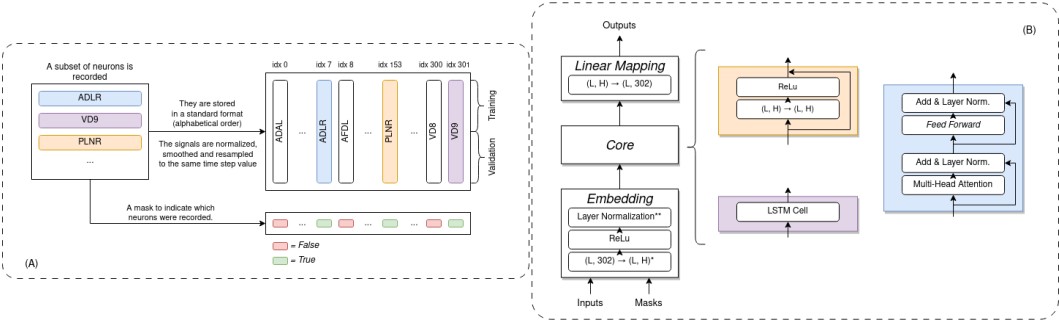

Figure 3: (A) Schematic representation of the data standardization process. (B) Schematic illustration of the three primary model architectures explored in this work: Linear (in yellow), LSTM (in purple), and Transformer (in blue). All models share a common foundational structure, shown on the left. For the experiments in this paper, we chose to use sequences of neural activity of length $L = 100$ time steps, corresponding to a duration of 100 seconds (i.e., $\Delta t = 1$s); however, the models are designed to handle sequences of arbitrary lengths. *In the Transformer architecture, a positional encoding is integrated within the embedding component. **Layer normalization is absent from the embedding component in the Transformer architecture.

## 2.3 BASELINE MODEL

**Baseline Model**. The baseline model posits that the next neural state will be identical to the current one, functioning as a naive predictor. This model serves as a reference point, particularly effective for random walk processes where the best prediction for the next step is the current state. In the context of neural activity data, this assumption challenges the neural network models to uncover and leverage more complex structures in the data beyond what is expected from a purely stochastic process.

**Baseline Loss**. The Mean Squared Error (MSE) calculated against this baseline often presents a deceptively simple yet challenging target for more sophisticated models, especially considering the

temporal coherence of neural signals. The baseline model's effectiveness underscores the necessity for more complex architectures to discern subtle patterns and dependencies within the data, which may not be captured by a simple random walk assumption.

## 2.4 TRAINING OBJECTIVE AND LOSS FUNCTION

**Training Objective**. The models are trained under a self-supervised paradigm, aiming to predict a one-timestep-shifted sequence of neural activity. Formally, this training objective can be expressed as minimizing the MSE between the predicted neural activity at time $t$ and the actual activity at time $t + \tau$, where $\tau$ is the lag and is set to 1 for immediate next-step prediction. The loss function further incorporates a boolean mask to ensure that only neurons with available data contribute to the loss computation, effectively focusing the learning process and the gradient updates. The mean-squared error (MSE) loss function with the boolean mask is defined as:

$$\mathcal{L}(\mathbf{X}, \hat{\mathbf{X}}, \mathbf{y}) = \frac{1}{\tau \times N \times L} \sum_{i=1}^{N} \sum_{t=0}^{L-1} \mathbf{y} \odot (\mathbf{X}_i(t) - \hat{\mathbf{X}}_i(t + \tau))^2,$$

where $\mathbf{X}_i(t)$ is the true activity of the $i^{th}$ neuron at time $t$, $\hat{\mathbf{X}}_i(t + \tau)$ is the predicted activity at time $t + \tau$, $\mathbf{y} \in \mathcal{R}^{302}$ is the boolean neuron or feature mask indicating the presence of data for neuron $i$, and $\tau$ is the timestep lag, which we set to 1 for immediate next-step prediction. Importantly, causality is ensured by only using past and present information to predict the future.

This self-supervised learning setup, also known as teacher forcing in training, guides models to predict an entire sequence shifted by one timestep, using the correct previous outputs. The boolean mask $\mathbf{y}$ is crucial as it adjusts the loss calculation to consider only the neurons with data, allowing for more efficient and effective learning, particularly when dealing with datasets that have varying availability of neuron measurements.

**Data Sampling and Model Evaluation**. We construct the training and validation sets by uniformly sampling 32 sequences of length $L = 100$ time steps from the first half of each worm's neural activity recording for training, and 16 sequences for validation from the second half. This methodology yields training and validation sets comprising $n \times 32$ and $n \times 16$ sequences, respectively, for a dataset of size $n$ worms. Data loaders for both training and validation utilize a batch size of 64.

**Training Protocol**. Models are trained up to a maximum of 500 epochs using the AdamW optimizer, with an initial learning rate of 0.001. A learning rate scheduler reduces the rate upon a validation loss plateau, with a decay factor of 0.1 and a patience of 10 epochs. Early stopping with a patience of 50 epochs is employed for efficiency.

## 3 RESULTS

### 3.1 DATA SCALING

#### 3.1.1 MIXED DATASET SCALING

**Objective.** To assess how increasing the amount of training data influences the model's predictive accuracy. The validation set is fixed, composed of neural activity data from all 284 worms, ensuring consistency in model evaluation.

**Approach.** We trained models on incrementally larger training sets created by sampling different numbers of worms from the pool of 284 worms (Figure 2). At each training set size, the models were evaluated against the same, fixed validation set, which is the largest compiled from the second half of neural recordings using all 284 worms across all experimental datasets.

**Results.** Figure 4 encapsulates the effect of training data volume on the MSE loss.

#### 3.1.2 INDIVIDUAL DATASET SCALING

**Objective.** To determine if models trained on datasets of varying sizes maintain consistent scaling properties when evaluated on a fixed validation set specific to each dataset.

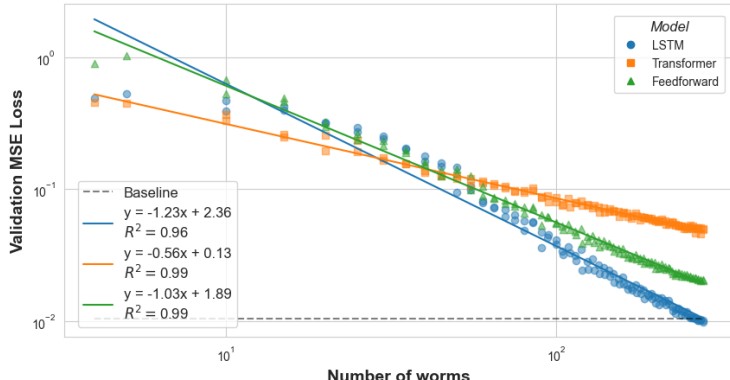

Figure 4: Validation MSE loss as a function of training data size for LSTM, Transformer, and Linear models (subplots A, B, and C, respectively). Each model's predictive accuracy improves with more training data, evaluated against a constant validation set comprising the complete pool of 284 worms. The dashed lines represent baseline MSE for comparison.

**Approach.** Utilizing the best model from the mixed dataset scaling experiment at each dataset size, we performed evaluations on a fixed validation set made from each of the individual experimental datasets (containing the validation sequences from all worms in that experimental dataset).

**Results.** Figure 5 portrays the similar scaling behaviors across models when validated against dataset-specific fixed validation sets.

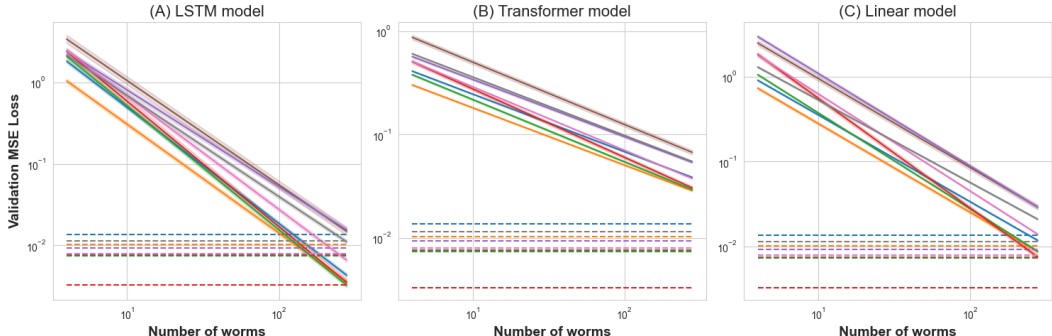

Figure 5: Individual dataset scaling behaviors displayed by LSTM, Transformer, and Linear models (panels A, B, and C). Consistent scaling is observed within model architectures when tested against the fixed validation sets corresponding to each dataset.

### 3.1.3 CROSS-DATASET GENERALIZATION

**Objective.** To explore models' abilities to generalize to independent datasets after being trained on a single dataset. Here again, a fixed validation set for each experimental dataset is used, comprising validation sequences from all worms from experimental dataset.

**Approach.** Each model was trained on the maximum sized training set of one of the original experimental datasets and tested against the maximum sized validation set of every the other experimental dataset.

**Results.** Figure 6 shows generalization performance, indicating models trained on more extensive datasets possess better adaptability.

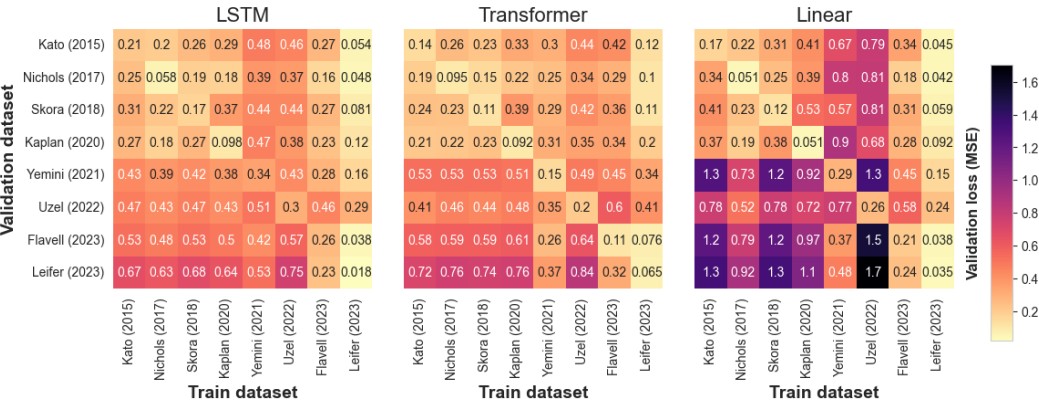

Figure 6: Heatmaps depicting the generalization capabilities of models across different experimental datasets (subplots for LSTM, Transformer, and Linear models). A fixed validation set, excluding the training dataset, was used to evaluate each model's ability to adapt to new neural dynamics, with models trained on larger datasets demonstrating enhanced performance.

## 3.2 HIDDEN SIZE SCALING

**Objective.** We investigated the influence of model complexity, as determined by the number of trainable parameters, on the performance of neural activity prediction in *C. elegans*.

**Approach.** Employing three neural network architectures—Linear, LSTM, and Transformer—with a unified architectural backbone (refer to Figure 3B), we modulated the hidden dimension size to study its impact on prediction accuracy. These modifications to the hidden layer width were systematically made across each model class, with training conducted on a consistent dataset, ensuring comparability.

**Results.** The experiment's results, depicted in Figure 7, indicate a non-linear relationship between the number of trainable parameters and the validation MSE. No fitted curve is imposed; instead, the data points exhibit a pronounced non-linear trend suggestive of an optimal parameter count for each model type. Beyond certain model sizes, further increases in parameters do not equate to improvements in prediction accuracy, highlighting the existence of an optimal model complexity for neural prediction in *C. elegans*.

## 4 DISCUSSION

The research presented herein aimed to decipher the scaling laws governing self-supervised learning models applied to neural activity prediction within *C. elegans*. Driven by the unique attributes of *C. elegans* as a model system, our work sought to understand the relationship between the volume of training data and the efficiency of different artificial neural network architectures in predicting neural states.

Our empirical results reveal a logarithmic decrease in mean squared error (MSE) as a function of increasing training data volume, a trend that held consistently across various experimental datasets. This suggests that the volume of data plays a crucial role in enhancing the predictive accuracy of neural activity models. Additionally, our investigation into the effects of model complexity indicated a non-linear relationship with prediction performance, with an observed optimal range of trainable parameters for each model type. Beyond this range, an increase in model size did not correspond to improved performance, suggesting the presence of an upper bound to the benefits of model complexity in this context.

The study's limitations include the challenge of determining the most appropriate model size for a given amount of data, as well as the exclusion of behavioral data, which could potentially provide additional contextual information for neural prediction. The latter represents a promising direction

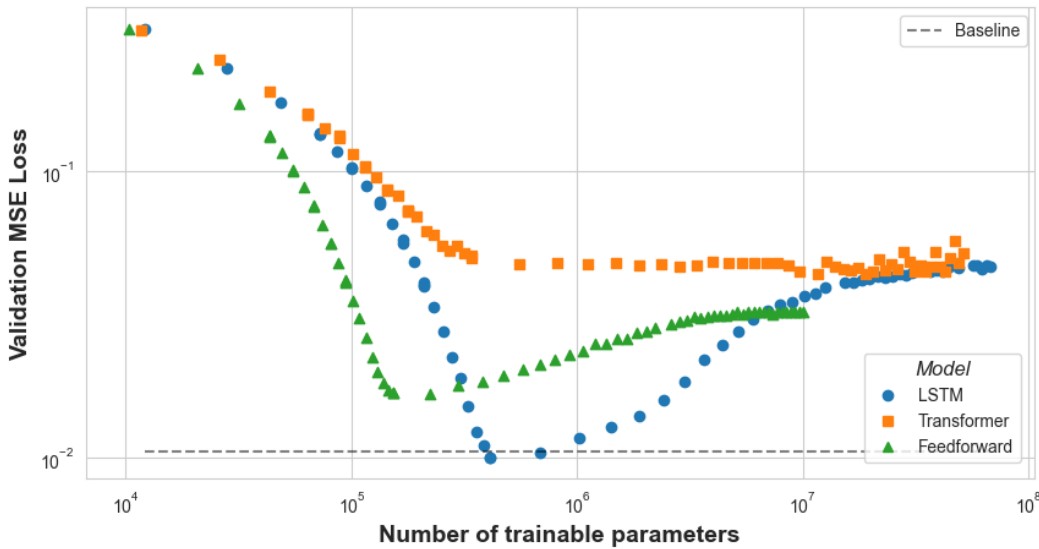

Figure 7: The relationship between the model's hidden size and validation MSE for LSTM (blue), Transformer (orange), and Feedforward (green) models. No quadratic fit is applied; the data points themselves reveal a non-linear relationship where performance peaks at an intermediate number of parameters, followed by a decline with further increases.

for future research to further elucidate the interplay between neural and behavioral data in predictive modeling.

In future work, we aim to refine our models to better account for the complexity of neural dynamics, exploring architectures that may effectively utilize larger datasets without incurring performance declines associated with excessive complexity. Additionally, integrating behavioral data into the predictive framework could potentially enhance the models' capabilities and yield insights into the neural basis of behavior. Extending our approach to more complex nervous systems could also provide valuable comparisons and contribute to the broader understanding of neural dynamics prediction.

Through these endeavors, we seek to advance the field of neural prediction, optimizing both the experimental and computational methodologies to better capture the essence of biological neural dynamics. The ultimate goal is to bridge the gap between model systems like *C. elegans* and more complex organisms, paving the way for models that can accurately reflect the intricacies of biological neural networks.

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

## A  APPENDIX

Table 1: Summary of publicly available *C. elegans* neural activity datasets. The table lists the associated publications, download sources, the format of the data files, the number of worms, and the mean number of labelled neurons versus all recorded neurons for each dataset.

| Paper Link | Database Link | Files w/ Data | Num. worms | Mean # neurons ID'd/recorded |
|---|---|---|---|---|
| tinyurl.com/Flavell2023 | wormwideweb.org | YYYY-MM-DD-*.json/h5 | 42 | 88/136 |
| tinyurl.com/Leifer2023 | osf.io/e2syt/ | exported_data.tar.gz | 103 | 69/122 |
| tinyurl.com/Yemini2021 | zenodo.org/records/3906530 | _Activity_OH*.mat | 49 | 110/125 |
| tinyurl.com/Uzel2022 | osf.io/3vkxn/ | Uzel_WT.mat | 6 | 50/138 |
| tinyurl.com/Kaplan2020a | osf.io/9nfhz/ | Neuron2019_Data_.mat | 19 | 36/114 |
| tinyurl.com/Skora2018 | osf.io/za3gt/ | WT_.mat | 12 | 46/129 |
| tinyurl.com/Nichols2017 | osf.io/kbf38/ | let.mat | 44 | 34/108 |
| tinyurl.com/Kato2015 | osf.io/2395t/ | WT_Stim.mat | 12 | 42/127 |

Table 2: Model Parameters Count

| A: Parameters Count for Hidden Size 512 | | |
|---|---|---|
| **Model** | **Hidden Size** | **Parameters Count** |
| LinearNN | 512 | 573742 |
| NetworkLSTM | 512 | 2412334 |
| NeuralTransformer | 512 | 1888046 |

| B: Matched Parameters Count (Approx. 0.574M) | | |
|---|---|---|
| **Model** | **Hidden Size** | **Parameters Count** |
| LinearNN | 512 | 573742 |
| NetworkLSTM | 232 | 573574 |
| NeuralTransformer | 262 | 573296 |

