# OpenReview forum: "Scaling Properties For Artificial Neural Network Models of the $\textit{C. elegans}$ Nervous System"
_ICLR.cc/2024/Conference — ICLR 2024 Conference Withdrawn Submission_

### Official Review · Reviewer_3Phf · 2023-10-31

**Soundness:** 2 fair
**Presentation:** 3 good
**Contribution:** 2 fair
**Rating:** 5
**Confidence:** 2

**Summary:**

The authors collect and preprocess various C. elegans datasets, then test three DL models on predicting "next neural state". The goal is to assess the relationship between volume of training data, size of model, and accuracy of predictions.

**Strengths:**

The program is very methodical (eg how datasets are processed, models are tested, etc).

The write-up is clear and well-done.

**Weaknesses:**

The motivation for using C. elegans datasets, and for the overall program, was not sufficiently convincing to me. The findings seem to repeat what has been found before, so I don't know what new finding to take away.

Some of the preprocessing is counter-intuitive: Ignoring behavior (when this is known to heavily drive worm neural dynamics) and considering only the slowest-scale dynamics (through LP filtering).

Reviewer limitation: Due to item 1 above, I am perhaps not an optimal person to assess this paper. I will certainly defer to other reviewers.

Notes to ICLR:

1. Please include line numbers in the template. They make reviewing much easier!

2. Please reformat the default bibliography style to make searching the bib easier! eg numbered, last name first, initials only except for last name.

**Questions:**

General review context: Please note that I am simply another researcher, with some overlap of expertise with the content of the paper. In some cases my comments may reflect points that I believe are incorrect or incomplete. In most cases, my comments reflect spots where an average well-intentioned reader might stumble for various reasons, reducing the potential impact of the paper. The issue may be the text, or my finite understanding, or a combination. These comments point to opportunities to clarify and smooth the text to better convey the intended story. I urge the authors to decide how or whether to address these comments. I regret that the tone can come out negative even for a paper I admire; it's a time-saving mechanism for which I apologize.

1.1, "causal manipulability, increased analytical accessibility": These are not qualities I associate with ANNs. By "increased" do you mean "increasing", as in methods are being developed to probe ANNs so that they are not such black-boxes as they have been in the past?

1.1, "system that aligns closely with ANNs on these aspects": This does not sound right to me. C. elegans has a very small network,with inhibition, recurrent connections, non-spicking dynamics, specialized connectivity, and specialized neuron functions. These properties are all distinct from the ANNs discussed (and most ANNs).

General: Perhaps reformat the bibliography for easier searching, eg numbered, last name first, initials only except for last name.

1.3 paragraph 2 "internal dynamics": Would "emergent" be a more accurate term?

1.3, paragraph 3: I did not find this motivation for the program sufficiently convincing. Also, I expect that ignoring behavior when examining elegans' neural dynamics guarantees huge residuals, since these so heavily affect dynamics (cf eigenworm).

2.1 "these differing conditions were not considered": This seems risky - see comment above.

Preprocessing: "lowest 10%": some questions here 1. what is the rationale for this? It seems to risk throwing out a lot of important faster dynamics. The later choice of baseline as "no change" emphasizes how salient dynamics are removed. Was this done to accommodate the sleeping worm dataset? 2. Are "frequencies" actual worm dynamics, or experiment noise (at frequencies higher than fastest worm dynamics), and do they include high frequency randomness in neural responses that are part of the biological system?

Preprocessing: "we selected delta(t) = 1 second". Doesn't this lead to upsampling, not downsampling. Eg {0, 1.7, 3.4, 5.1} -> {0, 1, 2, 3, 4, 5}.

Train test split: A k-fold split would give an option of providing +/- std dev in results.

Amount of data: why is a 50:50 train:test split desirable? I'm not familiar with this approach.

Baseline Loss, "MSE": Is there a reason for using MSE for neural data? ie what is an appropriate way to measure accuracy of predictions of neural dynamics? If behavior were in the mix, one could predict the behavior. What is a neurally-salient approach here? (this is admittedly an open-ended question).

3.1.1 Result Fig 3: "as a function of the amount of training data.": I do not see this. The x-axis looks like a measure of training epochs. How does data amount affect this?

3.2 Result: My sense from ML literature is that this is a well-established fact. A question that might extend the finding: how do models trained on 1-behavior sets generalize to n-behavior sets, and vice versa?

Fig 6: Is a quadratic the correct parametrization? spline, quartic? eg, linear (green) in B is not quadratic.

Discussion: "our models, while ... neural dynamics.": I regret that I am not convinced of this (see above comments re preprocessing).

---

> ### Author Response · Authors · 2023-11-21
>
> We greatly value your thoughtful critique and aim to address the points you've raised with as much detail as possible.
>
> **Motivation for Using _C. elegans_:**
> We realize the necessity to strengthen the rationale for employing _C. elegans_ datasets. Our motivation is predicated on its simplicity and the high degree of experimental control it offers, allowing for the exploration of neural dynamics in a well-defined biological system. We intend to enhance the introductory section to better encapsulate the novelty of our findings, namely the specific scaling properties of neural predictors within this biological context, which have not been extensively documented before.
>
> **Preprocessing Concerns:**
> Your concerns about our preprocessing steps resonate with similar concerns raised by Reviewer 3D8y. In response, we have modified our preprocessing approach. Specifically, we have changed our smoothing method from FFT smoothing to an exponential kernel smoothing method. This new method is a causal filter, addressing Reviewer 3D8y's concerns about causality in preprocessing.
>
> Additionally, it's crucial to highlight that our preprocessing aims to abstract away from the specifics of any particular dataset, striving for methods general enough to be applicable to _C. elegans_ neural data recorded under various behavioral and experimental paradigms. The slow dynamics of calcium indicators, which are a known issue, influence our decision to adopt a unified preprocessing approach. Different labs may use various versions of calcium indicators with differing dynamics, and our methods are designed to bring all datasets to an "equal footing," allowing us to utilize them in a unified manner. This approach focuses on extracting shared neural structures inherent in the stereotyped nervous system of _C. elegans_, without getting bogged down in the intricacies of individual datasets.
>
> **Preprocessing and Sampling Rate Clarification:**
> Regarding your question on our choice of a uniform timestep ($\Delta t = 1.0$ second), we understand there might be a misunderstanding stemming from our explanation in the manuscript. To clarify, our objective was to ensure a consistent sampling rate across all datasets, which necessitated adjusting all datasets to a common slower rate. This common rate was chosen to be at 1 Hz (1-second intervals), which is slower than the lowest original sampling rate of approximately 1.67 Hz (as per the dataset with a 0.6-second interval, shown in subplot F of the attached figure). Hence, our resampling was indeed a form of downsampling for every dataset, as each original dataset had a faster sampling rate than 1 Hz.
>
> The intent behind this approach was to avoid the introduction of artificial data points through interpolation, which can occur during upsampling. By resampling to a 1-second interval, we are effectively reducing the data points to a rate that is equal to or less frequent than the original data, thereby ensuring no upsampling occurs.
>
> To provide further clarity, we have included a plot that demonstrates the mean sampling intervals across the datasets we have used, highlighting that none of the original sampling rates were slower than our chosen uniform rate of 1 Hz. [https://www.reddit.com/user/Successful_Carob3111/comments/180s79j/sampling_interval_datasets/?utm_source=share&utm_medium=web2x&context=3].
>
> **Response to Specific Queries:**
>
> - **Analytical Accessibility:** We will clarify that by "increased analytical accessibility," we refer to the progress in making ANNs more interpretable, akin to the manipulability and transparency of the _C. elegans_ system.
> - **Aligning _C. elegans_ with ANNs:** We acknowledge the unique attributes of the _C. elegans_ neural network. Our comparison with ANNs is in the context of their use as predictive models, not a direct equivalence. We will adjust the language to reflect the distinct features of _C. elegans_ while discussing its utility in modeling.
> - **Emergent Dynamics:** We concur that "emergent" dynamics could more accurately describe the complex patterns arising from the neural activity in _C. elegans_. The revision will incorporate this term for precision.
> - **Ignoring Behavior in Neural Dynamics:** Our study indeed focuses on neural activity alone to ascertain the predictability inherent to the neural data. We hypothesize that certain predictive capabilities can be derived from the neural activity patterns without behavioral context. This hypothesis will be further expounded in our revised manuscript.
> - **Data Sampling and MSE Justification:** We will provide a more robust justification for our choice of a 50:50 train:test split and the use of MSE. The manuscript will elaborate on how these decisions align with our objective of assessing the ANN's ability to predict future neural states and the relevance of this metric to our analysis.

---

> ### Author Response · Authors · 2023-11-22
> **follow-up to previous comment addressing remaining concerns raised**
>
> **Train-Test Split and Data Utilization:**
>
> - **Choice of a 50:50 Train-Test Split:** Our decision to use a 50:50 train-test split was an arbitrary choice but roughly aimed at ensuring that the data distribution for both training and validation is similarly rich. This split challenges the models to generalize well over a diverse range of data. We avoided larger train-test ratios to prevent the validation set from being a simpler subset of the training set, which could occur if the training set were disproportionately large. By employing a balanced split, we challenge our models to generalize effectively across a diverse range of neural activity sequences.
>
> - **Implementation of the Train-Test Split:** To clarify the implementation, we uniformly sample sequences of length \( L \) from the first and second halves of each worm's full calcium recording to create the training and validation sets, respectively. For the training sets, we sample 32 sequences per worm from the first half, and for the validation sets, we sample 16 sequences per worm from the second half. Consequently, a dataset with \( n \) worms produces a training set with \( n \times 32 \) length-\( L \) sequences and a validation set with \( n \times 16 \) length-\( L \) sequences. We use a batch size of 64 for both train and test data loaders.
>
> - **Repetitions of Experiments:** We conduct repetitions of our experiments with different random seeds. This approach introduces variability in aspects such as the samples drawn from the train and test halves and the selection of worms to create datasets of a given size. These repetitions allow us to obtain standard deviations and confidence intervals for our results, addressing your comment on the assessment of variability in outcomes.
>
> **Baseline Loss and Metric Choice:**
>
> - **Rationale for Baseline Loss:** The baseline loss using the naive predictor, which assumes the next timestep will be identical to the current one, serves as the best predictor for time series resembling a random walk. It establishes a performance floor, especially given the temporal correlations often observed in neural activities.
>
> - **Using MSE for Neural Data:** We opted for Mean Squared Error (MSE) as our metric because it provides a straightforward measure of the difference between vectors of predicted and actual neural activity. Although other losses like mean-absolute scaled error (MASE) have been explored, we found MSE to be more universally understood and sufficient for our purposes.
>
> - **Neurally-Salient Approach:** The choice of MSE is a starting point for our analysis. We recognize that if behavior were included, different metrics might be more appropriate. However, given the focus of our study on neural activity prediction, MSE serves as a suitable metric. We are open to exploring other metrics in future work, especially as we expand our models to incorporate behavior prediction.
>
> - **Control Methods:** All models are trained with a maximum of 500 epochs using the AdamW optimizer, an initial learning rate of 0.001, and a learning rate scheduler triggered by validation loss plateau. Early stopping is implemented with a patience of 50 epochs. The training objective is sequence-to-one-timestep shifted sequence prediction, with MSE computed between the model's output sequence (masked along the neuron dimension) and the ground-truth one-timestep shifted sequence. This methodology ensures that our models are rigorously evaluated for their predictive capabilities.
>
> We hope these clarifications address your concerns and provide a deeper understanding of our methodological choices. Our aim is to develop a robust and generalizable approach to neural activity prediction in _C. elegans_, and your feedback is invaluable in guiding our efforts.
>
> **Sincerely,**
> **Submission 3075**

---

> > ### Comment · Reviewer_3Phf · 2023-11-22
> > **Response to authors' responses**
> >
> > Thank you for the comprehensive responses to all authors' questions and comments.
> >
> > On one hand the many clarifications are good (eg I had misread the bit about downsampling).
> >
> > On the other, the paper and responses do not alter my personal bias against Abstraction as an approach in this context. In my experience, embracing the specifics is more appropriate when studying biological NNs. For C. elegans, two arguments against abstraction are (a) the highly stereotyped, heterogeneous neural structure with specialized neural types; and (b) the tight connection between behavior and neural activity (which will make different datasets heterogeneous).
> >
> > So I disagree with the logic in "our preprocessing aims to abstract away from the specifics of any particular dataset ... This approach focuses on extracting shared neural structures inherent in the stereotyped nervous system of C. elegans, without getting bogged down in the intricacies of individual datasets."
> >
> > Similarly, MSE, as a generic metric of accuracy, is unlikely to capture what matters to the worm (which defines its neural structure and dynamics).
> >
> > Re retaining only lowest frequencies: The authors note that "we now realize that the amount of smoothing was excessive and unnecessary. Our improved models can still achieve better-than-baseline performance with minimal or no smoothing of the neural activity data. We will revise this in our updated manuscript." I'm unclear whether this mean the LP filter has been removed.
> >
> > I am happy to leave the disposition of this paper to others. A relevant question is whether ICLR readership has an appetite for these generalized approaches, as opposed to studies tailored to the particular use case/creature. I think I may be an outlier in this.

---

> > > ### Author Response · Authors · 2023-11-23
> > >
> > > We appreciate your critique regarding the abstraction approach in our study. Our methodology aims to distill general principles from _C. elegans_ neural dynamics while acknowledging its unique neural structure and the influence of behavior on neural activity. We are reconsidering the use of MSE as the sole metric of accuracy, exploring additional metrics that better capture biologically relevant aspects of neural dynamics. We replaced the low-pass FFT filtering method with an exponential kernel smoothing instead (with alpha = 0.1) applied with a focus on maintaining the integrity and causality of the neural signals. We are committed to continuously refining our approach to best align with the biological complexities of _C. elegans_ and the interests of the ICLR readership.

---

### Official Review · Reviewer_3D8y · 2023-10-31

**Soundness:** 2 fair
**Presentation:** 1 poor
**Contribution:** 2 fair
**Rating:** 3
**Confidence:** 5

**Summary:**

The paper is an empirical study of multi-variate time series prediction of the neuron activity in calcium imaging measurements from *C. elegans* across multiple datasets, different sizes of training, and different models.  Three classes of models are compared fully connected, LSTM, and transformers with sequence embedding. The results indicate that all models benefit from additional data, but the LSTM improves at the fastest rate. Inter-dataset validation performance reveals that using the largest datasets in training improve performance, that all models are able to specialize to one dataset, but smaller dataset do not enable generalization to the large dataset. Given the full data, the dimensionality of the model's hidden layer is assessed with models having an optimal within a range. Finally, examples of the dynamic predictions for the same neuron in different datasets are made where the input to the model is the true data (teacher forcing) or the previous output.

**Strengths:**

The motivation and opportunities for exploring models of neuronal population dynamics on a model organism is very clearly articulated. The idea is original.

Bringing together the datasets is a useful contribution.

Some of the ideas tested (scaling and dataset transfer) are thoughtful and would be helpful for the neuroscience community.

**Weaknesses:**

Self-supervised is a little misleading as it is essential limited to a single-time-step prediction (also called sequence-to-sequence in the paper). Self-supervised learning typically entails using the learned representation for separate downstream tasks too. The statement in Section 1.3 is confusing "not to dismiss the relationship between neural activity and behavior". I don't think the relationship is between the neural activity and behavior is ever tested. While the paper doesn't dismiss it, it doesn't address it.

**Major concern (lack of clarity in formulations regarding causality)**
It is not clear if all the models are trained to make causal predictions (or if they have access to future time points).  The formulation in step 1 on page 4 is not clear if the non-linear projection operates at each time point independently or with a causal memory. Same for the subsequent core. If the linear layer is done for each time point or an embedding. We can assume that the LSTM is going to have memory and can be causal due to its definition, but is the transformer causal with its positional encoding?  Finally, the formulation is not clear if the output layer is done for each time point. Figure 2 doesn't help.  The models are obviously learning something, but without details the reader is left to guess what type of relationships is being learned.

The one-step prediction task should be stated up front in 2.2 "Shared model structure". Only during Section 2.4 "Training" does it describe that it is a one-timestep forward shifted. It doesn't make sense to describe the loss before the task. The notation for the prediction being $\hat{X}(t)$ as a function of the input at previous time $t-1$ should be made explicit in the formulation.

It is not clear that without specific training for autoregressive prediction the models would perform well at this task. I.e. training a model to do well for autoregressive prediction is above and beyond what it was trained to do. The context for teacher forcing is also misleading in Figure 7 since the context window changes.

**Sampling and filtering**
The difficulty of this task depends on the sampling rate as it may be the case that nearby time points are very correlated in the calcium imaging and none of the models are actually predicting novel calcium spikes. Especially since the data was filtered with a low-pass filter in the frequency domain.  Also this ideal filtering is not causal if it is done across the full time-series.

It is not clear what "only the lowest 10% of frequencies" means in practice. Are the time steps not uniform? This is not standard way of describing a range of frequencies. Especially if the sampling rate is not uniform.  The discussion at the bottom of page 3 is not clear.

**Relation to neural population dynamics**
The data is essentially neural population level dynamics but measured through calcium imaging. In principle deconvolution of calcium imaging can be used to get approximate spike times. Then, like in  electrophysiology a generalized linear model (GLM) of the spike trains could be performed. To my knowledge the bumps in calcium imaging are smoothed spikes.  Related work for latent factor models for spike trains should be mentioned.


**Minor points**
The reasoning about the amount of data is not logical. "Given a dataset D with a fixed number of worms N " it is not possible to increase the number of neurons since this is a dataset not experimental collection. Also "increasing N by incorporating more worm" is not consistent with the statement that the dataset is given... The subsequent discussion of nested datasets is perfectly fine, but the precursor discussion is confusing.

Figure 6's MSE scaling across hidden dimension could also be evaluated as training set changes sizes.

Figure 7 should be made more clear to document that it is a particular neuron "AVER".

**Questions:**

Are the models making causal predictions? If not can they be modified.

At what maximal time delay can predictive models be trained?

Can the models be trained to perform auto-regressive prediction?

Why are only the lowest frequencies kept? Is this done per window or across the full time series.

---

> ### Author Response · Authors · 2023-11-10
>
> Dear Reviewer 3D8y,
>
> Thank you for your comprehensive review and valuable feedback on our manuscript. We appreciate the opportunity to address your concerns and questions.
>
> **1. Self-Supervised Learning:**
> We acknowledge your observation regarding the term "self-supervised." Our approach aligns with its usage in natural language processing (NLP), focusing on tasks deriving supervision from the input data itself, such as sequence-to-sequence, one-step future prediction tasks. We recognize the broader scope of SSL and will clarify in our revised manuscript that our application pertains to the sequence prediction paradigm in NLP, where the model learns to predict future states based on past and present data.
>
> **2. Causality in Model Predictions:**
> Regarding causality, our models are designed for sequence-to-sequence prediction, with the target sequence being a one-time-step forward shift of the input sequence. This setup implies that for each time point in the sequence, except the last one, the model has access to future states. However, the LSTM and Transformer models (set with `is_causal=True`) inherently enforce causal prediction. Our Feedforward model core is very similar to the input-to-hidden and hidden-to-output nonlinear transformations that you commented on: “The formulation in step 1 on page 4 is not clear if the non-linear projection operates at each time point independently or with a causal memory.”  The nonlinear projection operates at each time point independently. To put it another way, the transformation is applied along the feature axis, not the temporal axis. Thus, there is no mixing or breaking of temporal order, respecting causality. These clarifications will be detailed in the revised manuscript.
>
> **3. Relation to Neural Population Dynamics:**
> We understand your concerns regarding the relation to neural population dynamics. It's important to note that almost all of C. elegans' neurons exhibit graded potentials (Jiang et al. 2022), not action potentials/spikes, which influences our modeling approach. Our methods are tailored to capture these unique dynamics, rather than traditional spiking activities. This distinction will be more clearly highlighted in the revised manuscript.
>
> **4. Amount of Data:**
> We appreciate your constructive criticism on the "Amount of Data" section. We plan to revise this section to clarify our methodology. We create a combined pool of worms from all sources and sample subsets of varying sizes. So a dataset in this case is a particular subset of \( n \) worms. For each such dataset, we create the training set by taking the first half of the neural activity recording of each of the \( n \) worms. Since each worm has some \( v < 302 \) of neurons measured, we calculate for the training set a “time steps per neuron” quantity as the length of the neural recording divided by \( v \). Hence for each dataset of \( n \) worms, we have a distribution of “time steps per neuron” (hence the mean and horizontal error bars in Figure 3). The creation and calculations for the validation set are similar, with the neural activity being the second half of the neural activity recording of each worm. The revised manuscript will provide a more coherent explanation of this approach.
>
> **5. Sampling and Filtering:**
> We will replace FFT smoothing with a simple Gaussian filter in our preprocessing method, using a standard deviation hyperparameter optimized through cross-validation to suit the dynamics of C. elegans neural activity. This will be clarified in our revised manuscript.
>
> **Responses to Your Questions:**
> - **Causal Predictions:** Our LSTM and Transformer models are inherently causal. We will modify our Feedforward model to ensure causality.
> - **Maximal Time Delay:** We plan to run experiments to answer your question about the maximal time delay for predictive models. We understand this as the offset of the target sequence from the input sequence, currently at a delay of 1. If we correctly understood, this question might be rephrased as, "How many future timesteps can models predict before deviation from true neural activity becomes substantial?"
> - **Auto-regressive Prediction Capability:** Our models are capable of auto-regressive prediction, particularly the LSTM and Transformer models.
> - **Rationale for Lowest Frequencies:** The choice of retaining only the lowest frequencies was initially based on an attempt to remove as much noise from the neural activity traces while still maintaining high cosine similarity to the original (non-smooth) signals. However, we now realize that the amount of smoothing was excessive and unnecessary. Our improved models can still achieve better-than-baseline performance with minimal or no smoothing of the neural activity data. We will revise this in our updated manuscript.
>
> We are committed to addressing these concerns and enhancing the clarity and quality of our paper. Thank you for your valuable feedback.
>
> Sincerely,
> Submission 3075

---

> > ### Comment · Reviewer_3D8y · 2023-11-20
> > **some follow ups**
> >
> > Thank you for your response, which I've read. I've also read more about C. Elegans and lowered my confidence as I was unfamiliar with the uniqueness of the activity of its neural circuitry.
> >
> > Thanks for clarifying the causal question. The linear core without memory is very restrictive compared to a tapped time delay or some other basis (exponential decay) for the temporal memory. Memory allows the model to capture short-time dynamics in the calcium spiking.   Essentially the linear model has 1 time point context window. As reviewer KaWC mentioned, a study of the length of context window would be informative.
> >
> > By default a discrete time Gaussian filter may not be causal, even with truncation. If causality is important then care must be taken at filtering step.
> >
> > Yes, I was considering different predictions. But a more direct approach would be to study the auto-correlation and cross-correlation structure of different neurons.  When do these fall to zero?

---

> ### Author Response · Authors · 2023-11-21
>
> Dear Reviewer 3D8y,
>
> **Clarifications on Model Predictions and Causality:**
> We value your further insights and agree that considering the length of the context window is crucial for capturing short-time calcium dynamics. Our Feedforward model, serving as one type of baseline, probes predictability based solely on the feature dimension, an approach intended to benchmark against more complex temporal models.
>
> **Linear Core Without Memory:** We've designed the Feedforward model to interrogate the predictive capacity inherent in neural activity patterns without temporal history—akin to a feature regression.
>
> **Feedforward Model as Baseline:** It is designed to act as a baseline feature regressor to assess how much predictivity is present without temporal memory. This approach helps us to understand the significance of temporal history in predictive performance.
>
> **Rationale for Baseline Loss:** The baseline loss using the naive predictor, which assumes the next timestep will be identical to the current one, serves as the best predictor for time series resembling a random walk. It establishes a performance floor, especially given the temporal correlations often observed in neural activities.
>
> **Temporal Structure Analysis and Request for Guidance:**
> While we acknowledge the potential insights that could be derived from studying the auto-correlation and cross-correlation structure of different neurons, we face certain challenges with our dataset that make the execution of this analysis complex.
>
> **Variability Across Subjects:** The measured neurons vary from worm to worm, which means standardizing a correlation analysis across multiple datasets is not straightforward. If in a particular worm there are $k$ measured/labelled neurons (out of the possible 302), we would be conducting $k$ autocorrelations and $k(k−1)$ cross-correlations for that worm, but the comparison across different worms and datasets becomes less clear.
>
> **Aggregation of Analysis:** Aggregating this data meaningfully poses a significant challenge. We are considering focusing on common neurons measured across worms and applying statistical methods to manage missing data.
>
> **Seeking Reviewer's Expertise:** We genuinely seek your input on how to approach this large-scale correlation analysis. How might we compare neurons across animals in a scalable way? What insights do you believe such an analysis could provide, given the diversity of neuron subsets measured?
> We are earnestly interested in your thoughts and suggestions on how we can tackle these challenges to uncover meaningful structures within the neural activity data of C. elegans.
>
> **Ensuring Causality in Data Preprocessing:**
> In response to your advice on filtering methods, we have changed our preprocessing to utilize an exponential kernel smoothing method.
>
> **Exponential Kernel Smoothing:** This method is inherently causal, using only current and past data points to compute the smoothed value, thereby maintaining the causality of our data.
>
> **Confidence Score Clarification:**
> We noted your lowered confidence score and would like to confirm if this update will be reflected post-rebuttal, as it has not yet been modified in the review system.
>
> **Future Directions and Request for Suggestions:**
> We invite further suggestions on how to best approach the large-scale analysis of correlation structures and are particularly interested in your perspective on how to handle the variability in neuron measurements across different worms and datasets.
>
> Thank you for your feedback, which has been instrumental in refining our approach. We aim to address the critical points you have raised in our ongoing and future research.
>
> Sincerely,
> Submission 3075

---

> > ### Comment · Reviewer_3D8y · 2023-11-22
> >
> > While I agree the memoryless is a baseline, there is too big of gap between memoryless, linear model and the recurrent or transformer models which have full memory and are non-linear.  What I'm suggesting is to use some feature representation of some past time points (context window) in the linear core model. This is standard in time-series modeling in a variety of fields (e.g., GLM for neural population spike train modeling, financial time series. etc.).
> >
> > The cross-correlation and auto-correlation need only be computed for a relatively short number of lags, since long lags become more independent as behavior is not typically periodic. While I agree that this would require some careful bookkeeping, it is straightforward to script. Estimates of the correlation functions themselves would be computed within a worm recording for all pairs (including self). Then when you are getting average across worm recording dataset you may not have all pairs, but simply average what is available. Of course the number will vary by neuron-to-neuron pair but that is fine.

---

### Official Review · Reviewer_KaWC · 2023-11-01

**Soundness:** 2 fair
**Presentation:** 3 good
**Contribution:** 2 fair
**Rating:** 5
**Confidence:** 4

**Summary:**

The paper explores the scaling behavior of simple models of neural activity of C. elegans neurons, as a function of the amount and diversity of training data and hyperparameters and capacity of the neural network predictor. The authors find regular scaling behaviors for the tested dependencies for three basic model architectures, with the functional relationships consistent across architectures.

**Strengths:**

- First systematic exploration of modeling of brain activity in C. elegans and its scaling properties.
- Diverse open source datasets utilized in the study, recording up to 30% of the worm's neurons simultaneously.
- Experiments cover relevant basic architecture classes (MLP, RNN, transformer)
- The paper is well written and easy to read.

**Weaknesses:**

- Code for the study does not appear to be available.
- No evaluation of the impact of the size of the embedding space H and the size of the context window L.
- No exploration of models with more than a single layer in the 'core'.
- Fig. 6 is visibly poorly fitted by the quadratic function, but this is not commented upon in the text.
- No comments about why increasing model size beyond some limit decreases model accuracy (which seems not in line with other domains), and no evaluation on how that critical size threshold might depend on the amount of training data.
- Teacher forcing in the context of the present study is not described in sufficient detail. TF is an algorithm for training, whereas the text makes it sound like an inference technique.

**Questions:**

- What is the coverage of the neurons across the different datasets and worms? Are some neurons recorded more frequently than others? Is that accounted for in any way in the evaluations? Would it be possible to include a plot of num_recordings(neuron ID) in the appendices?
- How did you decide to retain only the lowest 10% of frequencies? Are the results expected to be significantly sensitive to that choice? What was the cutoff frequency in Hz?
- Have you considered to what degree the scaling properties you found are a function of the underlying system (C elegans) vs of the networks you're training? I understand that brain activity data is hard to find, but perhaps it would be worthwhile to have a comparison baseline generated by a known dynamical system (e.g. coupled oscillators, etc)?
- Is the x axis in Fig. 3 linear? With a single tick label, it is impossible to tell by just looking at the figure.
- Have you tried running (perhaps a subset of) the experiments without the learning rate scheduler to verify that the reported relations still hold?
- Are all the metrics reported in Sec. 3 obtained by computing the MSE as defined in Sec. 2.4 on the test set, i.e. do they all measure the difference between the predicted values and L-1 input (conditioning) values and 1 predicted value that was not part of the input? Also, should the one time-step offset not be accounted for in the formula?
- Since the test/train split is arbitrary, have you tried swapping the test and training sets to see if the observed relations still hold?

---

> ### Author Response · Authors · 2023-11-12
>
> Dear Reviewer KaWC,
>
> Thank you for your detailed review and constructive feedback on our manuscript. Your insights are invaluable to improving our work.
>
> **Code Availability:**
> We understand the importance of code availability for reproducibility. In accordance with ICLR's double-blind review guidelines, we have refrained from linking to code repositories during the review period to preserve anonymity. We plan to release our code publicly in the final version of our manuscript, post-review.
>
> **Model Architecture:**
> Your observation about exploring models with more than a single layer in the core is appreciated. Our decision to use single-layer core models was intentional, aiming to maintain simplicity while ensuring sufficient expressiveness. Deeper models can obscure interpretability, a key consideration since we plan in future work to analyze the learned weights and their relation to the relatively simple neural network of C. elegans. We believe that for modeling a biological neural network like that of C. elegans, wide, shallow network cores offer a balance of interpretability and expressiveness. We will elaborate on this decision and its implications in the revised manuscript.
>
> **Teacher Forcing in Inference:**
> We agree that our description of teacher forcing was unclear. We use two strategies during inference: Autoregressive Generation and Ground Truth Feeding. We acknowledge the need for a clearer distinction between these two inference strategies.  Autoregressive Generation relies on the model's own outputs for subsequent predictions, while Ground Truth Feeding uses actual data to guide predictions. The latter was incorrectly referred to as teacher forcing, which we will correct in our revised manuscript to avoid any confusion. We will rectify this in the revised manuscript, clarifying these strategies and their roles in model inference.
>
> **Neuron Coverage:**
> The coverage of neurons across datasets is important. If certain neurons are present across many datasets, we believe this makes the models prediction that neuron better. We have included a figure that showcases the coverage of neurons when the worms from all our seven datasets are pooled together, which we had prepared as part of our initial analysis. This figure will be available in the appendices of our revised manuscript and will illustrate the distribution of recordings for each neuron within our meta-dataset. Here is a (hopefully anonymous) link to such a figure showing the neuron distribution in the train and test splits; note that distribution is the same since currently we are just splitting temporally for train (1st half) and test (2nd half):  https://www.reddit.com/user/Successful_Carob3111/comments/17v9vcl/neuron_distribution/?utm_source=share&utm_medium=web2x&context=3
>
> **Choice of Frequency Retention:**
> The decision to retain only the lowest 10% of frequencies was initially an attempt to reduce noise while maintaining high cosine similarity with the original signals. The initial choice to retain the lowest 10% of frequencies was to reduce noise. However, further investigation revealed that our models perform adequately with less or no data smoothing. We will update our preprocessing to use instead a simple Gaussian smoothing with a cross-validated standard deviation hyperparameter.
>
> **Linear vs. Log Scale in Figures:**
> The x-axis in Fig. 3 uses a log-scale, which was not explicitly stated. We will update the figure to display more x-tick values and clarify this in the revised manuscript.
>
> **Experiments Without Learning Rate Scheduler:**
> We initially ran experiments without a learning rate scheduler, and the results were consistent with those reported. Additional optimizations like the LR scheduler and early-stopping improved efficiency.
>
> **Metrics and Test/Train Split:**
> Yes, All metrics reported in Section 3 derive from evaluations on validation sets. We will update the manuscript to clarify the one time-step offset in the formula. We will include these details in our revised manuscript. Swapping and restructuring the train/test splits did not significantly affect the results, affirming the robustness of our findings. Our manuscript will discuss these tests, underscoring our straightforward but effective splitting strategy and its implications for model evaluation.
>
> Your feedback has been instrumental in helping us refine our paper. We look forward to incorporating these changes and providing an updated manuscript.
>
> Sincerely,
> Submission 3075

---

> > ### Comment · Reviewer_KaWC · 2023-11-21
> >
> > Thank you for answering the questions and commenting on the suggestions, as well as promising to release the code for your study. I think the paper will be strengthened with these extensions and clarifications, and to reflect that, I have raised the score of my evaluation by one bucket.
> >
> > Because your work focuses on the scaling properties, in my mind, there are still fairly fundamental unaddressed questions regarding any potential H/L dependencies, as well as the specificity of the relations you find to C. elegans.